# New Insights into the Estimation of Reproduction Numbers during an Epidemic

**DOI:** 10.3390/vaccines10111788

**Published:** 2022-10-25

**Authors:** Giovanni Sebastiani, Ilaria Spassiani

**Affiliations:** 1Istituto per le Applicazioni del Calcolo Mauro Picone, Consiglio Nazionale delle Ricerche, Via dei Taurini 19, 00185 Rome, Italy; 2Mathematics Department “Guido Castelnuovo”, Sapienza University of Rome, Piazzale Aldo Moro 5, 00185 Rome, Italy; 3Department of Mathematics and Statistics, University of Tromsø, H. Hansens veg 18, 9019 Tromsø, Norway; 4Istituto Nazionale di Geofisica e Vulcanologia (INGV), Via di Vigna Murata 605, 00143 Rome, Italy

**Keywords:** reproduction number, epidemic evolution, SARS-CoV-2, estimation techniques, mathematical analysis

## Abstract

In this paper, we deal with the problem of estimating the reproduction number $R_{t}$ during an epidemic, as it represents one of the most used indicators to study and control this phenomenon. In particular, we focus on two issues. First, to estimate $R_{t}$, we consider the use of positive test case data as an alternative to the first symptoms data, which are typically used. We both theoretically and empirically study the relationship between the two approaches. Second, we modify a method for estimating $R_{t}$ during an epidemic that is widely used by public institutions in several countries worldwide. Our procedure is not affected by the problems deriving from the hypothesis of $R_{t}$ local constancy, which is assumed in the standard approach. We illustrate the results obtained by applying the proposed methodologies to real and simulated SARS-CoV-2 datasets. In both cases, we also apply some specific methods to reduce systematic and random errors affecting the data. Our results show that the $R_{t}$ during an epidemic can be estimated by using the positive test data, and that our estimator outperforms the standard estimator that makes use of the first symptoms data. It is hoped that the techniques proposed here could help in the study and control of epidemics, particularly the current SARS-CoV-2 pandemic.

## 1. Introduction

To study and control an epidemic such as the current SARS-CoV-2 pandemic, positive test case data ($Y_{p\;\;t}$) and first symptoms data ($Y_{f\;\;s}$) sequences are commonly used. These data are independently gathered and respectively describe the number of positive tests officially registered each day and the number of patients exhibiting symptoms on the first day. The data of the first kind are commonly communicated by the media, while those of the second kind are elaborated to compute the instantaneous reproduction number $R_{t}$ during the evolution of an epidemic [1,2].

Both kinds of data are affected by errors. Positive test data have an intrinsic problem: the result of a test is typically associated with the day of its official registration with the authorities. It would be more appropriate to consider the day of the test. In addition, tests have false negative and false positive outcomes. However, molecular tests are more commonly used, with a typical sensitivity of 90%; these tests are different from the rapid antigenic tests, whose sensitivity largely varies and can be much lower. First symptoms data have two main problems. First, not all positive patients are able to contextualize their symptoms into a time frame, which reduces the sample size. Second, these data are continuously reported to the medical authorities, and the number of first symptoms on each day is updated until a certain time, after which it stabilizes.

The study and estimation of the reproduction numbers during an epidemic is an active field of research. A large number of different methodologies, approaches, and practical applications have been developed.

Fraser [3] proposed to use a slight modification of the classical Kermack–McKendrick transmission model to estimate the $R_{t}$ [4] based on factors that influence susceptibility and infectiousness. In addition, Fraser [3] specifically focused on households as they are considered a fundamental unit of transmission for several infections that are directly transmitted. He then analysed a model of transmission within and between households and developed a method to estimate their reproduction number.

Specific to the SARS-CoV-2 pandemic are in-host disease models, which have largely been analysed due to the relevant role played by co-morbidities and co-treatments (e.g., [5,6]), as well as methodologies which tackle the issue of asymptomatic cases. For example, Zhao et al. [7] performed a combined estimation of the generation interval and incubation period, and then inferred a pre-symptomatic transmission proportion and latent period.

Several proposed methodologies are based on simple or generalized compartmental models that combine deterministic and stochastic components, thus allowing for a consideration of various external variables such as quarantine, self-isolation, social distancing, or infection profiles [8,9,10,11]. Nevertheless, public institutions such as the Istituto Superiore di Sanità (ISS) in Italy do not use such models to estimate $R_{t}$. They instead use a far simpler data-driven approach that is based on the simple statistics of first symptoms onset data [12], on which we have focused part of our work. This approach is preferred as it does not formulate any hypothesis on the mechanisms of virus transmission. One main problem with compartment models is the assumption of homogeneity: it is very unrealistic, even at the minimum geographical level used in Italy, to assume that each individual can infect every other person. In addition, there are problems related to the number of parameters and their estimation.

The method described by Cori et al. [12] was implemented for the computation of $R_{t}$, both in the R software [13] and as a Microsoft Excel spreadsheet ([14] and documentation in SI of [12]), which was named EpiEstim. In [15], an extensive review was performed on articles that used EpiEstim to describe modified approaches. It was reported that the method of [12], possibly modified, had been used to compute $R_{t}$ in more than 280 papers, most of which were for SARS-CoV-2. Even if the data used were concerned with the incidence of first symptoms and not of infection, the $R_{t}$ computed from them by the method in [12] could provide useful quantitative information about the evolution of the SARS-CoV-2 pandemic.

First symptoms data satisfy an equation involving a convolution integral [12]:(1)$$Y\left(t\right)=R_{t}\left(t\right)\left(Y*f\right)\left(t\right)=R_{t}\left(t\right)\int_{0}^{t}Y\left(s\right)f\left(t-s\right)\;ds,$$
where $Y\left(\cdot \right)=Y_{f\;\;s}\left(\cdot \right)$, the symbol ∗ indicates the convolution, and f(·) is a proper kernel function representing the probability density function (PDF) of the serial interval, which is the time between the onset of the first symptoms and the infection that had caused them (e.g., [16]). Equation (Equation 1) is the basis for the estimation of the instantaneous reproduction number, but we point out that it can be rigorously derived only when $R_{t}$ is constant along time and there are no asymptomatic individuals. Nevertheless, it is commonly used in real applications, either as a definition of $R_{t}$ or as a basic equation to derive an estimator for it.

We stress that in practice, the convolution integral in Equation (Equation 1) is numerically approximated by a linear combination of data, and this is also performed here in the applications. However, in some theoretical calculations, we will consider the convolution as an integral. For simplicity, we will use the same symbol.

In an ideal condition, first symptoms data are replaced by the infection data [17]. However, it is obvious that data on the last type of events are hard to collect. In fact, the time of first symptoms can be quantified, although with errors, while it is very often not possible to identify the infection time. We notice that Equation (Equation 1) becomes an eigenfunction problem if $R_{t}$ is assumed to be constant over time. In principle, the eigenfunctions could be used as a base of functions to describe the $R_{t}$ sequence.

Usually, in the literature, the kernel function *f* is modelled by a Gamma PDF:$$f\left(y\right):=f_{C}\left(y\right)=\frac{1}{\Gamma \left(k\right)\theta^{\kappa }}y^{k-1}e^{-y/\theta },$$
where *k* and θ are the shape and the scale (positive) parameters, respectively, and Γ indicates the Gamma function. Based on a limited set of epidemic data, these parameters were estimated for the SARS-Cov-2 pandemic in Italy by [18] to be 1.87 (shape) and 3.57 (scale), and this is indeed the model currently adopted by the dedicated Italian Governmental Institution to estimate the reproduction number (see https://www.iss.it/coronavirus; for the theoretical specifications, see https://www.iss.it/coronavirus/-/asset_publisher/1SRKHcCJJQ7E/content/faq-sul-calcolo-del-rt). We stress that other parameter values are used in different countries (e.g., [19,20]).

In this paper, we focus on one of the most widely used methods to estimate $R_{t}$ during an epidemic [12]. It is developed within a Bayesian statistical framework [21], which uses both the likelihood of the data $Y\left(t\right)=Y_{f\;\;s}\left(t\right)$ for t=1,…,T and a *prior* model for the temporal $R_{t}$ sequence. As it will be seen in the next section, by following this approach, we can derive the expression for an estimator ${\hat{R}}_{t}\left(t\right)$ in a closed form, that is:(2)$${\hat{R}}_{t}\left(t\right)=\frac{a+\sum_{s\in I_{t}}Y\left(s\right)}{\frac{1}{b}+\sum_{s\in I_{t}}\left(Y*f_{C}\right)\left(s\right)},$$
where $Y*f_{C}$ denotes the convolution discretized by a linear combination of data, and $I_{t}$ is a temporal interval containing time *t*. Intuition suggests that the choice of $I_{t}$ to be symmetrically centred in *t* can be adopted to minimize a possible bias of the estimator. To obtain this formula, it is assumed that the instantaneous reproduction number is constant within $I_{t}$. This hypothesis allows for a reduction of the influence of random errors in measurements during the estimation of $R_{t}$. Mathematically, this is obtained by the two “averaging” terms appearing both in the numerator and the denominator of Equation (Equation 2). However, we stress that this procedure may induce systematic errors. In fact, the slope of the estimated $R_{t}$ is often reduced (in absolute value) with respect to that of the underlying “true” curve. Finally, the constancy over $I_{t}$ is independently assumed from time to time without any guarantee of consistency. The strategy adopted here aims to overcome the problem induced by the “averaging” procedure of the standard approach to estimate $R_{t}$. Hence, we do not assume $R_{t}$ to be constant over the interval $I_{t}$.

This paper first describes a new procedure that is closely related to the one in [12], which is currently used in Italy to estimate $R_{t}$. The method is simple, both to understand and to implement, and it has been applied to both simulated and real SARS-CoV-2 Italian data at different time intervals. However, it could be easily adapted to other contexts worldwide. Secondly, we illustrate the theoretical and empirical results we found on the relationship between the $R_{t}$ estimation based on first symptoms and that based on positive test data. In Section 2, we first give some details of the method in [12], and we describe the methodologies developed in our work. In Section 3, we illustrate the results obtained by applying the proposed methodologies to both synthetic and real SARS-CoV-2 Italian data. For comparison, the estimates from the standard method in [12] are included as well. Finally, the results are discussed in Section 4.

## 2. Mathematical Models and Methods

In this section, we first provide some details about the method in [12] to derive Equation (Equation 2). Secondly, we illustrate how it is possible to reduce the errors of the data, both for first symptoms and positive test sequences. Then, we focus on the estimation of $R_{t}$ based on positive test data as a valid alternative to the currently used first symptoms data. At the end of this section, we describe the proposed modification to the method in [12] to estimate the instantaneous reproduction number $R_{t}$ during an epidemic.

### 2.1. A Standard Method to Estimate $R_{t}$

In the method by [12], the data $Y\left(t\right)=Y_{f\;\;s}\left(t\right)$ for t=1,…,T are assumed to be independent and identically distributed (i.i.d.) as a Poisson distribution, and therefore the expression for their likelihood in terms of the parameter vector $R_{t}$ is as follows: (3)$$L_{Y}\left(R_{t}\right)=\prod_{s=1}^{T}\frac{{\left[R_{t}\left(s\right)\cdot \left(Y*f_{C}\right)\left(s\right)\right]}^{Y(s)}\;\;e^{-R_{t}\left(s\right)\cdot \left(Y*f_{C}\right)\left(s\right)}}{Y(s)!},$$
where we used Equation (Equation 1) to approximate the expected value of Y(s). The *prior* model on the instantaneous reproduction number in the days of the time interval considered is assumed to be the product of the PDFs, which are all equal to a Gamma distribution with shape a=1 and scale b=5 (see [12]). Those two probabilistic models are then combined by means of the Bayes theorem to obtain the *posterior* PDF, that is: (4)$$P\left(R_{t}\;|\;Y\right)\propto \prod_{s=1}^{T}{\left[R_{t}\left(s\right)\cdot \left(Y*f_{C}\right)\left(s\right)\right]}^{Y(s)}\;\;e^{-R_{t}\left(s\right)\cdot \left(Y*f_{C}\right)\left(s\right)}\;R_{t}{\left(s\right)}^{a-1}e^{-R_{t}\left(s\right)/b}.$$

Equation (Equation 3) factorizes because of the independence of the different components of $R_{t}$; therefore, the inference can be performed independently component by component. To do that, we consider the *posterior* marginal distribution density function of the variable $Q=R_{t}\left(t\right)$ given by the following equation:(5)$$P_{Q}\left(q\right)\propto q^{Y(t)+a-1}\exp\left\{-q\left[\frac{1}{b}+\left(Y*f_{C}\right)\left(t\right)\right]\right\},$$
which is a Gamma distribution with shape Y(t)+a and scale ${\left[\frac{1}{b}+\left(Y*f_{C}\right)\left(t\right)\right]}^{-1}$ as parameters. The mean estimator is then adopted. We recall that the mean of a Gamma-distributed random variable is the product between the shape and the scale. Then, in our case, we obtain the following equation:(6)$${\hat{R}}_{t}\left(t\right)=\frac{a+Y(t)}{\frac{1}{b}+\left(Y*f_{C}\right)\left(t\right)}\;t=1,\ldots ,T.$$

When the order of magnitude of the measured data is at least in the hundreds, the prior parameters (a,b) in Equation (Equation 6) can be neglected. In fact, this situation typically happens in the case of a very large population studied in an active phase of the pandemic, long after its beginning.

The above estimator ${\hat{R}}_{t}\left(t\right)$ is, however, affected by random errors induced by those of the first symptoms sequence. The contribution of the denominator in (Equation 6) to the errors is lower than that of the numerator because of the presence of the convolution, discretized by a linear combination of data. To further reduce the influence of the errors on $R_{t}\left(t\right)$, it is assumed that the instantaneous reproduction number is constant within an interval $I_{t}$. Then, the marginal distribution density of $R_{t}$ becomes a Gamma PDF with shape $a+\sum_{s\in I_{t}}Y\left(s\right)$ and scale ${\left[\frac{1}{b}+\sum_{s\in I_{t}}\left(Y*f_{C}\right)\left(s\right)\right]}^{-1}$. With abuse of notation, the corresponding estimator ${\hat{R}}_{t}\left(t\right)$ is, in this case, given by the right-hand side of Equation (Equation 2). Since summation now also appears in the numerator and twice in the denominator, the influence of random errors is further reduced, which is reflected in a smoother $R_{t}$ curve. However, as pointed out in the Introduction, systematic errors may appear.

### 2.2. Data Error Reduction

Often, in an epidemic such as the current SARS-CoV-2 pandemic, both positive test and first symptoms sequences, besides random fluctuations, contain variations that are approximately described by a weekly oscillating component, with a local minimum on Mondays and a local maximum on Wednesdays/Thursdays. This is more evident for positive test data, as some steps involved to produce them are dependent on the day of the week. Therefore, we process the measured sequences to produce a version of them with reduced periodic and random distortions, as often performed in the literature (e.g., [22,23]). To do that, we model the data as the sum of two components. The first consists of a non-parametric Nadaraya–Watson linear combination of data [24,25]:(7)$$\tilde{Y}\left(t\right)=\frac{1}{F}\sum_{s=1}^{T}Y\left(s\right)K\left(\frac{t-s}{\gamma }\right),\;F=\sum_{s=1}^{T}K\left(\frac{t-s}{\gamma }\right).$$
where Y(1),…,Y(T) are the measurements. In the above formulas, K(·) is the *kernel* function, and the positive parameter γ is the *bandwidth* [26]. The kernel is typically modelled by a standard Gaussian PDF:$$K\left(x\right)=1/\sqrt{2\pi }\;e^{-\frac{x^{2}}{2}},$$
and this is also performed here. The second component of the model that we propose for the data is parametric: it consists of a sinusoidal function with a period of 7 days, which is multiplied for each time by an “envelope” function. The specific choice of the latter depends on the data considered (see Section 3.1).

The whole set of model parameters consists of the bandwidth of the non-parametric component and the coefficients of the second one. To select the optimal values for all the parameters, we proceed as follows. For each element of the bandwidth in a finite set of selected values, we initially compute the first component directly from the data with Equation (Equation 7). Then, correspondingly, we estimate the parameters of the periodic component by optimizing the fit of the complete model to the data. Among the estimated models, we finally select the one which gives the best fit to the data. At the end of this procedure, when dealing with the standard estimator for $R_{t}$, we use the data obtained by subtracting the optimal periodic component from the measured data. For simplicity, we hereafter use the same notation Y(1),…,Y(T) for this “filtered” sequence. In contrast, to estimate $R_{t}$ through the proposed method, we use the non-parametric component $\tilde{Y}\left(1\right),\ldots ,\tilde{Y}\left(T\right)$ of the best model.

### 2.3. Estimation of $R_{t}$ from Positive Test Data

As intuition suggests, there exists a relationship between the positive test and the first symptoms sequences. In fact, a contribution to the incidence $Y_{p\;\;t}\left(t\right)$ of a positive test registered at time *t* is given by the integral $\int_{0}^{t}Y_{f\;\;s}\left(\tau \right)\;P\left[Y_{p\;\;t}\left(t\right)|Y_{f\;\;s}\left(\tau \right)\right]\;d\tau$ of the incidence $Y_{f\;\;s}\left(\tau \right)$ of the first symptoms outcomes that occur at any time τ<t, weighted by the conditional probability density $P[Y_{p\;\;t}\left(t\right)|Y_{f\;\;s}\left(\tau \right)]$. Of course, we also have to take into account the contribution to the incidence $Y_{p\;\;t}\left(t\right)$ of the subjects that are not able to localize their first symptoms outcomes. Assuming that this last contribution is approximately proportional to the total incidence $Y_{p\;\;t}\left(t\right)$, it follows that
$$Y_{p\;\;t}\left(t\right)\propto \int_{0}^{t}Y_{f\;\;s}\left(\tau \right)\;P\left[Y_{p\;\;t}\left(t\right)|Y_{f\;\;s}\left(\tau \right)\right]\;d\tau .$$By assuming that the mechanisms for sampling, executing tests, and registering the results remain the same along time, the above conditional probability only depends on the temporal difference t−τ. Therefore, the sequence of the positive test is obtained by the convolution of the first symptoms sequence with an appropriate kernel function g(·):(8)$$Y_{p\;\;t}\left(t\right)=\int_{0}^{t}Y_{f\;\;s}\left(\tau \right)g\left(t-\tau \right)\;d\tau .$$

We notice that the function g(·) no longer integrates to 1, as it incorporates a constant larger than 1 due to the presence of the subjects that cannot identify the day of their first symptoms.

Let us now replace the expression for the first symptoms sequence in the previous Equation (Equation 8) with a convolution integral given in (Equation 1), where we explicitly set $f:=f_{C}$, which is the kernel introduced by [18]:$$Y_{p\;\;t}\left(t\right)=\int_{0}^{t}\left[R_{t}\left(\tau \right)\int_{0}^{\tau }Y_{f\;\;s}\left(x\right)f_{C}\left(\tau -x\right)\;dx\right]g\left(t-\tau \right)\;d\tau .$$Since, in practice, the support of the kernel g(·) is 14 days, we can write the following equation:$$\begin{array}{cc} Y_{p\;\;t}\left(t\right) & =\int_{t-14}^{t}\left[R_{t}\left(\tau \right)\int_{0}^{\tau }Y_{f\;\;s}\left(x\right)f_{C}\left(\tau -x\right)\;dx\right]g\left(t-\tau \right)\;d\tau \\ \; & =R_{t}\left(\;\bar{t}\;\right)\int_{t-14}^{t}\int_{0}^{\tau }Y_{f\;\;s}\left(x\right)f_{C}\left(\tau -x\right)\;dx\;g\left(t-\tau \right)\;d\tau \\ \; & =R_{t}\left(\;\bar{t}\;\right)\int_{0}^{t}\left(Y_{f\;\;s}*f_{C}\right)\left(\tau \right)\cdot g\left(t-\tau \right)\;d\tau =R_{t}\left(\;\bar{t}\;\right)\cdot \left[\left(Y_{f\;\;s}*f_{C}\right)*g\right]\left(t\right)\\ \; & =R_{t}\left(\;\bar{t}\;\right)\cdot \left[\left(Y_{f\;\;s}*g\right)*f_{C}\right]\left(t\right)=R_{t}\left(\;\bar{t}\;\right)\cdot \left(Y_{p\;\;t}*f_{C}\right)\left(t\right), \end{array}$$
where we have applied the general version of the mean value theorem, and $\bar{t}$ is a certain time in the interval (t−14,t). In general, $\bar{t}$ is an unknown function $\bar{t}\left(t\right)$ of time *t*. However, we can assume that the mechanisms producing the first symptoms outcomes in a subset of infected subjects, and all those that involve testing in this subpopulation, remain identical during a reasonably limited time period. Of course, this is not true in the very first stages of the pandemic. Therefore, in a limited temporal interval, the whole composite phenomenon is “homogeneous” along time. For such kinds of phenomena, given a generic time *t*, the point $\bar{t}\left(t\right)$ where $R_{t}$ is calculated in the final expression of $Y_{p\;\;t}\left(t\right)$ above, must have the same distance from *t* independently of the absolute location of time *t* along the temporal axis. It then follows that $\bar{t}\left(t\right)=t-\delta$, where δ is a constant, and we have the equation below:(9)$$Y_{p\;\;t}\left(t\right)=R_{t}\left(t-\delta \right)\;\left(Y_{p\;\;t}*f_{C}\right)\left(t\right)=R_{t}\left(t-\delta \right)\int_{0}^{t}Y_{p\;\;t}\left(s\right)f_{C}\left(t-s\right)\;ds.$$

We notice that Equation (Equation 9) is identical to Equation (Equation 1), but the $R_{t}$ is shifted by the constant quantity δ. This is intuitive, as the process of the positive test sequence ideally resembles that of the first symptoms, with a delay. The same is true between the latter and the infection sequence. However, we recall that the infection sequence is unknown, and that in fact, the kernel in [18] was estimated by using a very limited sample of primary infection data (in tens or hundreds).

We also want to empirically verify Equations (Equation 8) and (Equation 9). We then come back to the kernel function *g* in Equation (Equation 8). It is usually modelled by a Gamma distribution density function with shape *k* and scale θ parameters. However, to verify Equation (Equation 8), we have to re-parametrize this model by replacing the scale parameter θ with the mean value μ=k·θ, which has a direct interpretation. We also consider an additional multiplicative parameter in the kernel, which allows us to account for the fact that the day of the first symptoms is only known for a fraction of all the patients that tested positive. The explicit formulation we consider for the kernel function is then given by the following equation:(10)$$g\left(t\right)=A{\left(t-t_{0}\right)}^{k-1}\exp\left\{-\frac{(t-t_{0})k}{\mu }\right\}.$$

The components of the parameter vector $\left(A,k,\mu ,t_{0}\right)$ of this model are estimated by minimizing the discrepancy between the measured sequence of the positive test incidence and its theoretical expression, which we recall is given by the discrete version of the convolution integral in (Equation 8). For the minimization, we use the simulated annealing algorithm with a geometric temperature schedule [27]. The initial value for the temperature is chosen as the empirical mean of an initial random exploration of the parameters state space. The value for the exponent of the geometric law is 0.999.

### 2.4. A New Estimator for the Reproduction Number Sequence

Similar to the standard method in [12], the method proposed here to estimate $R_{t}$ during an epidemic is also based on Equation (Equation 1). As we explained before, in the standard approach, when estimating $R_{t}$ on day *t*, one assumes that its value is constant within an interval $I_{t}$ centred on *t*. In this way, the estimator ${\hat{R}}_{t}$ on day *t* will depend on the data in the whole $I_{t}$ (see Equation (Equation 2)). This is done in order to increase the regularity of the ${\hat{R}}_{t}$ curve. However, in order to reach the same goal with our approach, we first compute the sequence of the non-parametric component $\tilde{Y}\left(1\right),\ldots ,\tilde{Y}\left(T\right)$ from the data, as described in Section 2.2. After that, we estimate the $R_{t}$ on day *t* by using Equation (Equation 6), thus obtaining the following equation:(11)$${\hat{R}}_{t}\left(t\right)=\frac{a+\tilde{Y}\left(t\right)}{\frac{1}{b}+\left(\tilde{Y}*f_{C}\right)\left(t\right)}.$$

We recall that the convolution in the denominator above is a discrete approximation of the convolution integral given by a linear combination of the sequence $\tilde{Y}\left(1\right),\ldots ,\tilde{Y}\left(T\right)$. Of course, we are aware that Equation (Equation 6) is theoretically derived under the assumption of the data’s independence at different times. However, this assumption is not valid here as we perform the first step in reducing the errors in the data.

## 3. Results

### 3.1. Real SARS-CoV-2 Italian Data

We start by considering a first set of SARS-CoV-2 Italian data for the first symptoms and a positive test, measured from 21 September until 20 November 2020. We notice that the date 14 September 2020 corresponds to the beginning of school activity in Italy; this induced a fast growth in the disease incidence, which started about two weeks later [28]. In Figure 1, we show both the first symptoms and the positive test measured data as well as the results of the methods to reduce their errors. As shown in this figure, the processed data are less affected by the presence of a one-week periodicity. The reduction of the errors is stronger for the positive test sequence. This is an expected result, firstly because the corresponding values are typically larger than those of the first symptoms. More importantly, the latter data have fewer factors that induce periodicity in the measurements. For the positive test sequence, the envelope chosen for the periodic component modelling the data (see Section 2.2) is proportional to the first component. This assumption is reasonable as the within-week variation is expected to be proportional to the mean value. This hypothesis is also supported by the empirical evidence given in Figure 2, where the standard deviation of the local fluctuations of the positive test sequence is shown as a function of the estimated first component of the model for $Y_{p\;\;t}\left(\cdot \right)$. To compute the standard deviation, we use the discrepancies between the measured positive test data and the estimated non-parametric component of their model, in slicing windows of 14 days centred on each point of the time interval. A degree 2 polynomial has been adopted. The optimal values for the parameters are obtained through an iterative least square procedure. We notice that in this case, we end up, in practice, with a linear relationship, which justifies the above choice for the envelope. However, this does not happen for data in another time interval, as we are going to see later on. The same procedure for computing the standard deviation and the same type of quadratic model are also adopted in the case of the first symptoms data (see Figure 3). In general, a quadratic model is adopted for the envelope in terms of the first component of the data model. Since the periodic distortions only appear in a sub-window of the time interval considered, we only applied the correction at that point. In Figure 1, the estimated first component of the data model is shown for both types of data. In addition to systematic errors, those that are random are also strongly reduced.

As explained before, the sequences of the first symptoms and positive test data can be linked to each other through a convolution relationship based on a suitable kernel (see Equation (Equation 8)). This is empirically verified by the results in Figure 4. We stress that this convolution link is a crucial element at the basis of the theoretical relationship between the estimate of the $R_{t}$ from the first symptoms sequence and the sequence corresponding to the positive test curve. In particular, the calculations in Section 2.3 show that the $R_{t}$ sequence, which is based on positive test data, can be obtained from analogous data based on the first symptoms by a translation of a suitable time δ.

This theoretical result is empirically verified on the real SARS-CoV-2 data considered here. By means of the standard approach, we estimated $R_{t,fs}$ from the first symptoms sequence and $R_{t,pt}$ from the positive test sequence. In both cases, we used the data obtained by subtracting the optimal periodic component from the raw measurements (see Section 2.2). We stress that by following the method used to compute and publicly diffuse the official $R_{t}$ values from the Italian ISS, the estimation temporal interval $I_{t}$ was 14 days long, centred on *t*: {t−7,…,t+6}. To find the optimal value for δ, we focused on the mean absolute difference between the $R_{t,pt}$ and $R_{t,fs}$ sequences in the time interval considered, which was performed after shifting backward the first sequence of a variable number of days, from 1 to 10. We then selected the δ that could minimize this difference, and we found an optimal value of 6 days. This choice corresponds to a good agreement between the two sequences after shifting, as shown in Figure 5. The value of the mean absolute difference in the time interval considered is 0.02.

We also applied the same analysis to a similar dataset in a different temporal interval, i.e., the period of one month starting on 7 December 2021, when the highly contagious Omicron variant of SARS-CoV-2 largely spread in Italy for the first time [29]. In addition to the differences due to the virus variant, the testing conditions were then quite different. The results we obtained are illustrated in Figure 6, from which we can draw the same conclusion as for the previous dataset considered. The optimal value we found for the shift δ was 4 days. This shorter delay is expected, since we are now considering a period in which the possibility for a person to be tested with reliable results is far easier than before. There is also the possibility of self-testing by means of easily purchasable antigenic tests, and a new, more reliable antigenic test (COI) that gives rapid results. For completeness, in Figure 7, we show the standard deviation of the local fluctuations of the positive test sequence as a function of the estimated first component of the model for $Y_{p\;\;t}\left(\cdot \right)$.

We now turn to the results obtained by the proposed method to evaluate the reproduction number $R_{t}$ during an epidemic for the first real dataset considered. They are illustrated in Figure 8, which also contains the results from the standard approach for comparison. As we can see, the $R_{t}$ curve from the proposed method is more regular than the curve from the standard approach. Around day 30, the differences between the two curves are small. This is not true for the first part of the interval, where the slope of the $R_{t}$ curve is high (until about day 15), because a systematic deviation appears: the estimate of the $R_{t}$ with the proposed approach shows a higher slope than that estimated in the standard case. In contrast to this latter approach, in the proposed one, we first reduced both the systematic and random errors in the data, and then we applied the estimator in (Equation 6). As an alternative, one could first apply the same formula to the data after reducing the systematic errors and then perform an arithmetic averaging of $R_{t}$ in the interval $I_{t}$. The result is shown in Figure 9, where it can be noticed that the proposed method outperforms this last procedure.

Confidence Intervals (CIs) for $R_{t}$ from both the proposed and standard methods are computed as follows. After estimating the two model components from the data, we generated a sample of *n* simulated signals first by adding these two sequences to each other, and then summing the resulting i.i.d. Gaussian noise with time-varying standard deviation, as shown in Figure 3. For each element of the data sample, we estimated the $R_{t}$ with the use of both the proposed and the standard methods. We stress that in order to make a fair comparison, the data processed by the standard approach were obtained from the simulated signals following a procedure the same as that used for the real data. For each of the two methods, we computed the 95% CI at each time *t* in the temporal interval considered. Using a sample of (n=100) independent replications, the maximum and mean values for the semi-amplitude of the $CI_{t}$ were respectively (0.07,0.1) for the proposed method, and (0.04,0.07) for the standard one. Since the posterior of $R_{t}$ for the standard method is a Gamma model, we used it to compute $CI_{t}$. However, in this specific case, the values of the $CI_{t}$ became substantially smaller than those above, and we therefore did not follow this procedure.

Similarly to what had been done before, we further validated the results of the comparison with the standard method by analysing the data in the second time interval, as above. In fact, due to the large spread of the Omicron variant, this period is also characterized by a high slope of the curve, thus allowing us to highlight the differences between the standard approach and our proposed method. The two estimated $R_{t}$ curves are shown in Figure 10. We can see that the $R_{t}$ from our method shows a slightly higher slope and higher peak, correctly retracing the result from the raw first symptoms data, represented in the figure by a dotted line.

Finally, the same results as those above have also been obtained from data in different countries. An example is given in Figure 11, where we consider SARS-CoV-2 data in New York City (NYC), measured from 26 October to 25 November 2020. We can also see that in this case, the $R_{t}$ from our proposed method shows a higher slope and a higher peak, occurring after the first two weeks of October. The peak could be due to the fact that at the beginning of October, NYC elementary, middle, and high schools began in-person learning.

### 3.2. Synthetic Epidemic Data

As anticipated before, to further validate the effectiveness of the proposed method from a quantitative/objective point of view, we generated synthetic epidemic data. We first obtained the synthetic data by modelling the incidence of first symptoms events by means of a pseudo-Gamma PDF (see (Equation 10)), with parameter vector (13289,22,83,0) plus an extra additive parameter C=551. This choice was made to obtain a “realistic” synthetic dataset, resembling the real data illustrated in panel a) of Figure 1. Given the regularity of the generated first symptoms events, we computed the “true” $R_{t}$ sequence by using Equation (Equation 6). We stress that in general, since the kernel by [18] has, in practice, a support of 28 days, in order to get the value of $R_{t}$ at any time, we need the data relative to the 28 days before it. An i.i.d. sample of the *n* signal was then simulated by adding an i.i.d. zero mean Gaussian noise to the standard deviation, growing as the same degree 2 polynomial model used for the first set of the real data.

Figure 12 shows the mean of the $R_{t}$ curves and its 95% CI estimated by the proposed method (n=100). The $CI_{t}$ is computed first by following the same procedure as that used for the real data, and then applying the Law of Large Numbers. The same figure shows the “true” $R_{t}$ curve, which is almost always contained in the 95% CI band. Analogously, Figure 13 contains the same result for the standard method. In this case, the “true” $R_{t}$ is not included in the 95% CI. Therefore, this provides additional evidences in favour of our procedure.

In addition to the simulated results just described, we compared the performances of the standard and the proposed methods by using another synthetic dataset. In this case, we started directly from the $R_{t}$ sequence. More precisely, we considered an $R_{t}$ that was constantly equal to 1 until a certain time t=0, after which it increased linearly with a slope α (ramp). Correspondingly, the sequence of the first symptoms data was explicitly derived by imposing that Equation (Equation 1) be fulfilled for the times in the interval considered (−∞,T], with T>0. Since $R_{t}\left(t\right)=1$ for t≤0, it is easy to verify that this implies that ${\left\{Y\left(t\right)\right\}}_{t=-\infty ,\ldots ,-1}\equiv Y\left(0\right)$. We then set Y(0)=1. For any time t=1,…,T, we obtain the following equation (see Appendix A):(12)$$Y\left(t\right)=\frac{R_{t}\left(t\right)\sum_{k=1}^{t-1}f_{C}\left(t-k\right)Y\left(k\right)}{1-R_{t}\left(t\right)f_{C}\left(0\right)}+\frac{R_{t}\left(t\right)\left[1-\sum_{x=0}^{t-1}f_{C}\left(x\right)\right]}{1-R_{t}\left(t\right)f_{C}\left(0\right)}.$$

Equation (Equation 12) can then be used recursively to compute the values of ${\left\{Y\left(t\right)\right\}}_{t=1,\ldots ,T}$. In panel (a) of Figure 14, the “true” $R_{t}$ sequence is shown. In panel (b) of the same figure, we plot instead the corresponding first symptoms sequence obtained by recursively applying Equation (Equation 12). Considering the noiseless case, from Equation (Equation 2) with $a=\frac{1}{b}=0$, we can explicitly compute the estimate of $R_{t}$ in the standard case. By looking at the results of both the real SARS-CoV-2 and the synthetic epidemic data of the first type, we notice that the bias is larger at the beginning of the interval where the $R_{t}$ has a high slope. Therefore, we focus on the bias $\hat{R_{t}}\left(0\right)-R_{t}\left(0\right)$ at time t=0. As shown in the Appendix A, the standard estimator $\hat{R_{t}}\left(0\right)$ is given by the following equation:(13)$$\begin{array}{cc} \hat{R_{t}}\left(0\right) & =R_{t}\left(0\right)+\frac{\sum_{i=1}^{6}\left[R_{t}\left(i\right)-R_{t}\left(0\right)\right]\left(Y*f_{C}\right)\left(i\right)}{\sum_{i=-7}^{6}\left(Y*f_{C}\right)\left(i\right)}=R_{t}\left(0\right)+\frac{\sum_{i=1}^{6}\left[R_{t}\left(i\right)-R_{t}\left(0\right)\right]\frac{Y(i)}{R_{t}\left(i\right)}}{\sum_{i=-7}^{6}\frac{Y(i)}{R_{t}\left(i\right)}}, \end{array}$$
where we used the basic Equation (Equation 1), and where $R_{t}\left(t\right)\equiv R_{t}\left(0\right)$ for t≤0. By inserting in Equation (Equation 13) the known values of the $R_{t}$ sequence and those of Y(·) calculated from the system Equation (Equation 12), the considered bias is obtained.

The corresponding bias for the proposed method is instead derived as follows:(14)$$Bias_{proposed}=\frac{\tilde{Y}\left(0\right)}{\left(\tilde{Y}*f_{C}\right)\left(0\right)}-R_{t}\left(0\right),$$
where the sequence $\tilde{Y}\left(\cdot \right)$ is obtained by applying the Nadaraya–Watson estimator (Equation 7) to the sequence Y(·), calculated from the system Equation (Equation 12). In Figure 15, we plotted the bias as a function of the slope α for both the standard and the proposed methods at day 21 (t=0). For the proposed method, we used three different (fixed) values of the bandwidth γ appearing in (Equation 7). The highest value of γ was close to the one (2.7) estimated from the real SARS-CoV-2 first symptoms Italian data. By looking at the figure, we notice that all the bias curves increase with α. The curve of the bias from the standard approach is above all the curves from the proposed method. Furthermore, by increasing the value of the fixed bandwidth γ, the curve of the proposed method gets closer to that of the standard approach.

## 4. Discussion and Conclusions

In this paper, we dealt with estimating the reproduction number $R_{t}$ during an epidemic. We first analysed, both theoretically and empirically, the relationship between the $R_{t}$ estimated from the first symptoms and the positive test data. Second, we modified the standard method by [12], which is widely used to estimate $R_{t}$ in several countries worldwide, including Italy: we did not rely on the hypothesis of the local constancy of $R_{t}$. To perform both tasks, we also developed a specific method to reduce errors in both the first symptoms and the positive test data.

We have proved, both theoretically and empirically, that the $R_{t}$ curve estimated from the positive test sequence is equal to the curve estimated from the first symptoms after shifting the last one forward by a temporal quantity δ. After a certain time, the value of δ may change, depending on some specific factors. For example, the average time between the first symptoms and a positive test may change under some conditions, e.g., the season. In addition, δ may change with the spatial location. In any case, the value of δ is *a-priori* unknown, and it must be estimated. Several approaches can be used. One could use a combined set of first symptoms and positive test data. After estimating $R_{t}$ from each of the two kinds of data, δ can be found by minimizing the discrepancy of the two $R_{t}$ curves (after shifting). Alternatively, one could rely on the meaning of this shift, interpreting it as the time delay Δt between the occurrence of the first symptoms and the positive results of the test. More precisely, from a dataset of patients where this delay is known for each individual, δ can be estimated as the median value of Δt. For example, in the case of the SARS-CoV-2 data for the Veneto region (Italy) during the first epidemic wave, the median value of the temporal delay is 6 days [30]. This value is equal to the one obtained by applying the first approach to the first real dataset used. We point out that instead of aiming at replacing the first symptoms data with the positive test data, one could use both of them to get two estimates of $R_{t}$, which would likely enhance the reliability of the results. In addition, estimating $R_{t}$ from positive test data has the advantage of being based on a far larger sample.

We notice that by the standard method, we cannot estimate $R_{t}$ closer than 6 days from the last measurement. This is also true when using positive test data. However, as we have seen, the estimated $R_{t}$ at any day is equal to the $R_{t}$ from the first symptoms process corresponding to δ days before it. For the value of δ estimated here (6 days), it follows that the last day in which we can estimate $R_{t}$ is 12 days before the last measurement. On the other hand, we notice that $R_{t}$ estimates based on first symptoms data are reliable until 16 days before the last measurement. In fact, these data are stable only approximately 10 days before the last measurement. Therefore, by using positive test data, we also gain 4 days, which increase to 6 for the second real dataset, as in this case δ is 4 days.

Regarding the new methodology for estimating $R_{t}$, the results obtained show that it outperforms the standard approach. In fact, in the regions with a high $R_{t}$ slope, the estimate from the latter approach has a lower slope as compared to the proposed one, while in the remaining part, there is good agreement. This is a real effect, as the pattern appears for both real and “realistic” synthetic first symptoms data. From the ramp synthetic data, we have seen that the bias increases with the slope α of the ramp. However, for the standard estimator, the bias is always larger than that of our method, and its rate of increase with α is greater for the standard method.

In the standard approach, the estimator is the mean of the posterior distribution (in most cases and in practice, the likelihood of the data), and one simultaneously imposes that $R_{t}$ is constant within two weeks. As usual, in this situation, the price for the induced regularity of the $R_{t}$ curve is a reduction of its slope in the region where it is large. Instead, in our method, we reduce the influence of the errors on the data by first smoothing them. Then, the estimate of $R_{t}$ is obtained by independently maximizing the posterior probability at each time. The estimates obtained are not affected by the problem above. This is true for both the first symptoms and the positive test data. It would be interesting to investigate if this happens in general or under certain conditions.

By applying our simple methodologies, either the standard method applied in Italy to compute $R_{t}$ for SARS-CoV-2 based on the first symptoms sequence is improved, or a valid alternative (or additional) approach is provided to the one using the first symptoms data when applying the standard method to positive test data. We notice that we also developed more advanced methods, e.g., the Bayesian approach with an a priori probability model on the temporal continuity of $R_{t}$, obtaining very similar results. However, the former approach is more difficult to understand, and its implementation is more complicated as it involves Markov Chain Monte Carlo simulation to perform statistical inference, which is also time-consuming. In addition, there is the problem of hyper-parameter estimation. We were able to cope with all these issues, but we wonder—why should one use a more complicated approach to get similar results to those derived from a far simpler one?

As a conclusion, the results illustrated in this paper suggest that the reproduction number $R_{t}$ during an epidemic can also be estimated by applying the standard estimator to positive test data, with some advantages. In addition, the new estimator we proposed for $R_{t}$ outperforms the standard one. We hope that the extensive application of these procedures to real situations, including the current SARS-CoV-2 pandemic, will become a common practice that could help in the study and control of epidemics.

## Figures and Tables

**Figure 1 vaccines-10-01788-f001:**
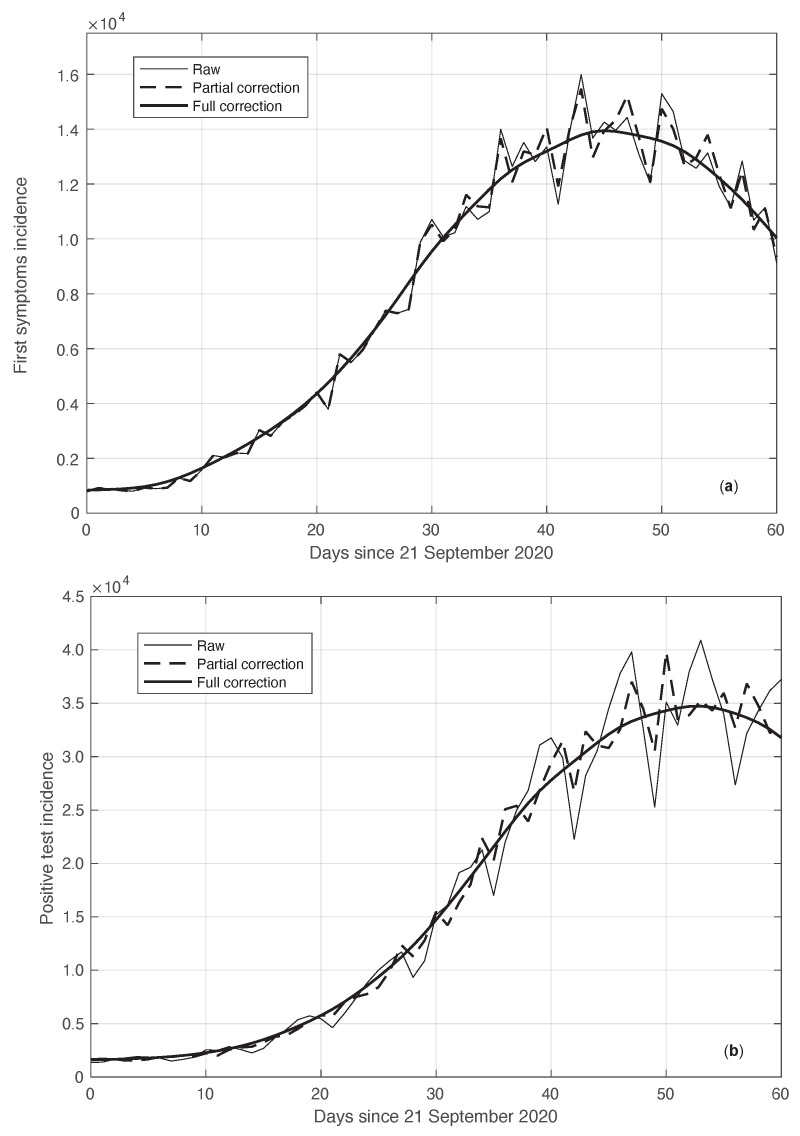
Real SARS-CoV-2 Italian incidence (number of new cases per day) from 21 September to 20 November 2020. In panels (**a**,**b**), the first symptoms and positive test data are respectively plotted. Measured data (thin continuous lines) are shown together with their corrected version from the one-week periodic component (thick dashed lines) and the non-parametric component of the model (thick continuous lines). See Section 2.2 for details.

**Figure 2 vaccines-10-01788-f002:**
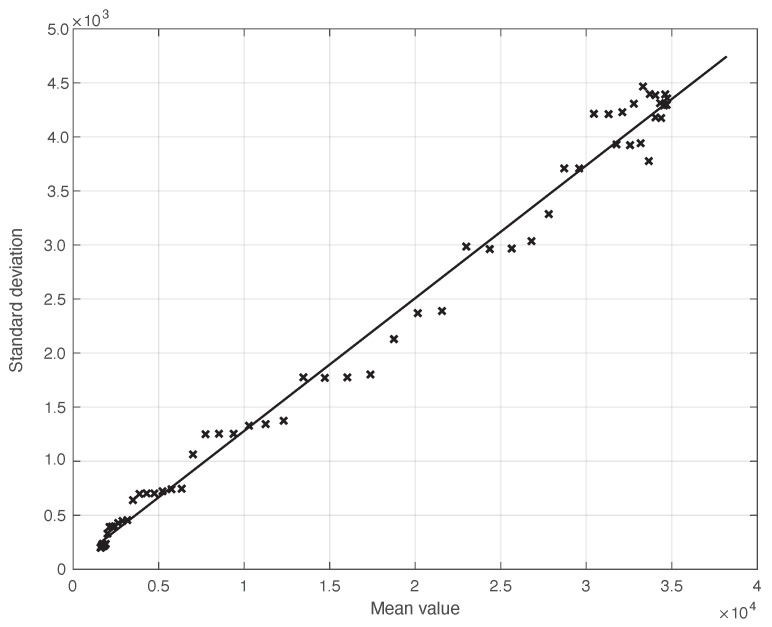
Standard deviation of the local fluctuations of a real SARS-CoV-2 Italian positive test sequence from 21 September to 20 November 2020, illustrated in Figure 1, as a function of their expected value. The continuous line represents the best fit with a degree 2 polynomial model, which in this case is reduced to a straight line.

**Figure 3 vaccines-10-01788-f003:**
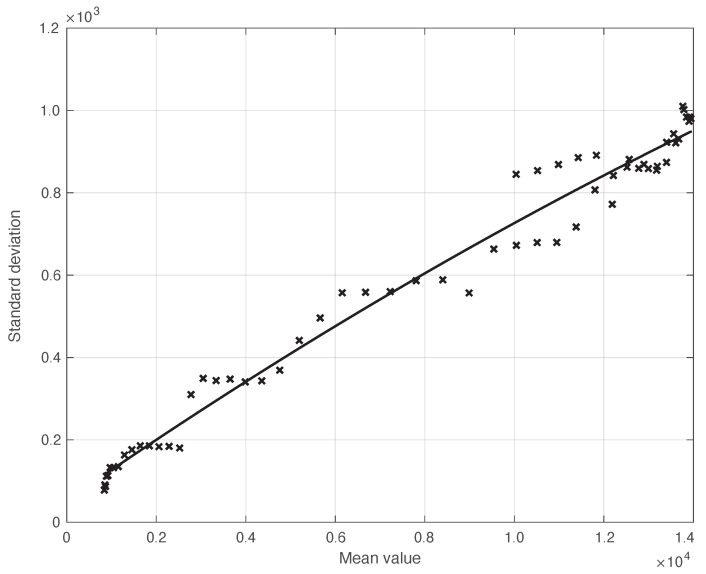
Standard deviation of the real SARS-CoV-2 Italian first symptoms sequence from 21 September to 20 November 2020, illustrated in Figure 1, as a function of their expected value. The continuous line represents the best fit with a degree 2 polynomial model.

**Figure 4 vaccines-10-01788-f004:**
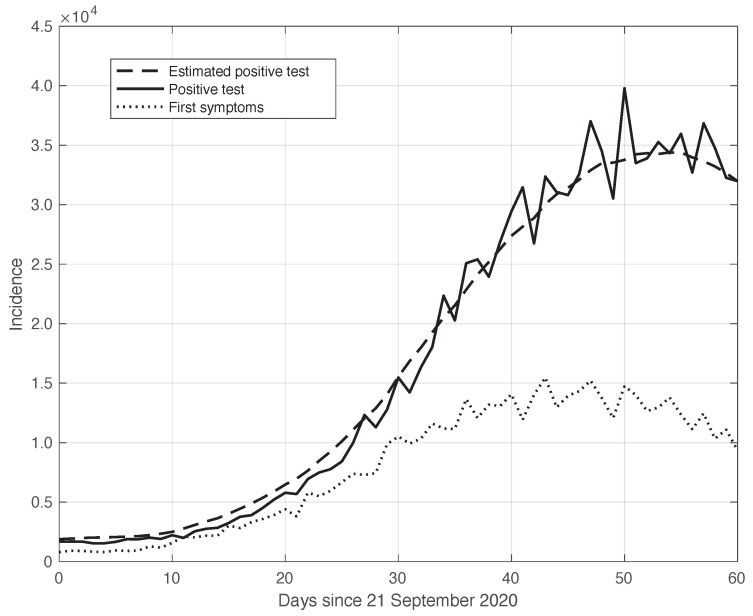
Relationship between the sequences of the positive test (continuous line) and the first symptoms (dotted line) relative to the real SARS-CoV-2 Italian data from 21 September to 20 November 2020, illustrated in Figure 1. The curves were obtained by correcting the data from a one-week periodic component (see Section 2.2). The dashed line shows the results of the convolution between the first symptoms sequence and the optimal kernel (see Section 2.3).

**Figure 5 vaccines-10-01788-f005:**
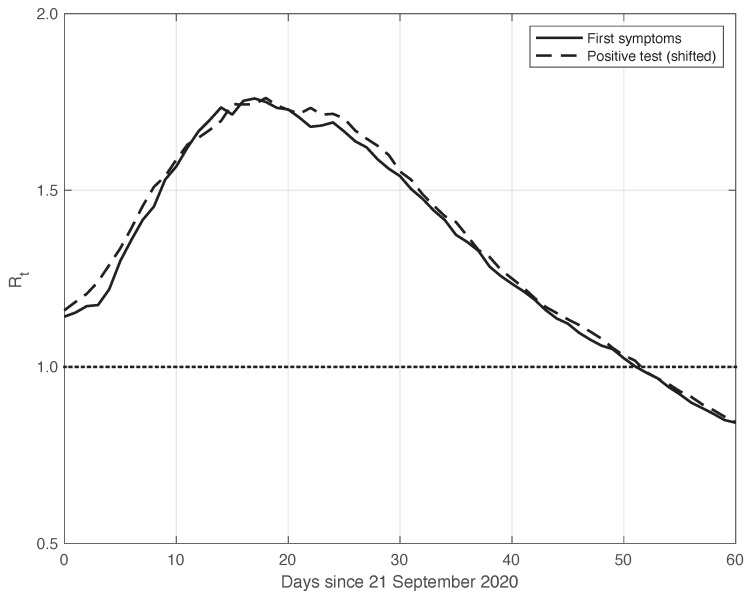
Sequence of the reproduction number $R_{t}$ estimated by the standard method for real SARS-CoV-2 Italian data from 21 September to 20 November 2020, illustrated in Figure 1. The continuous and dashed lines refer to the first symptoms and the shifted positive test sequences, respectively. The estimated optimal shift is 6 days. The dotted line represents the threshold for the epidemic to spread or die out.

**Figure 6 vaccines-10-01788-f006:**
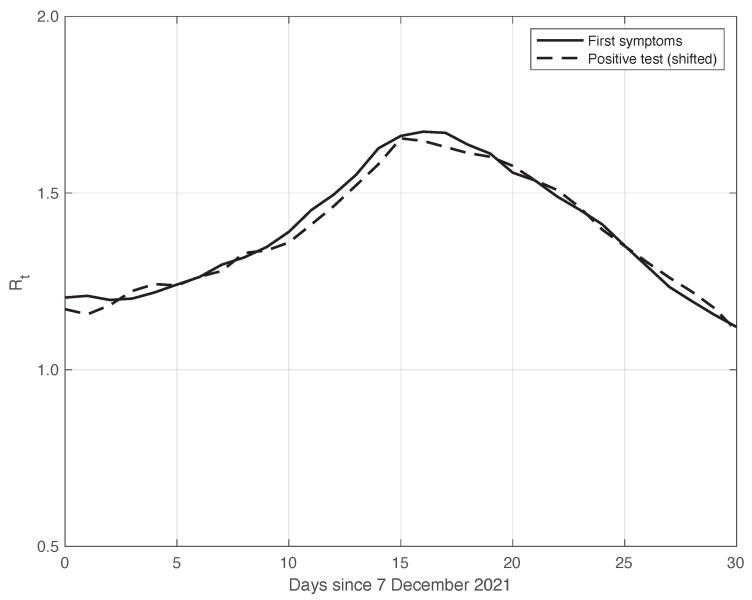
Sequence of the reproduction number $R_{t}$ estimated by the standard method for real SARS-CoV-2 Italian data from 7 December 2021 to 6 January 2022. The continuous and dashed lines refer to the first symptoms and the shifted positive test sequences, respectively. The estimated optimal shift is 4 days.

**Figure 7 vaccines-10-01788-f007:**
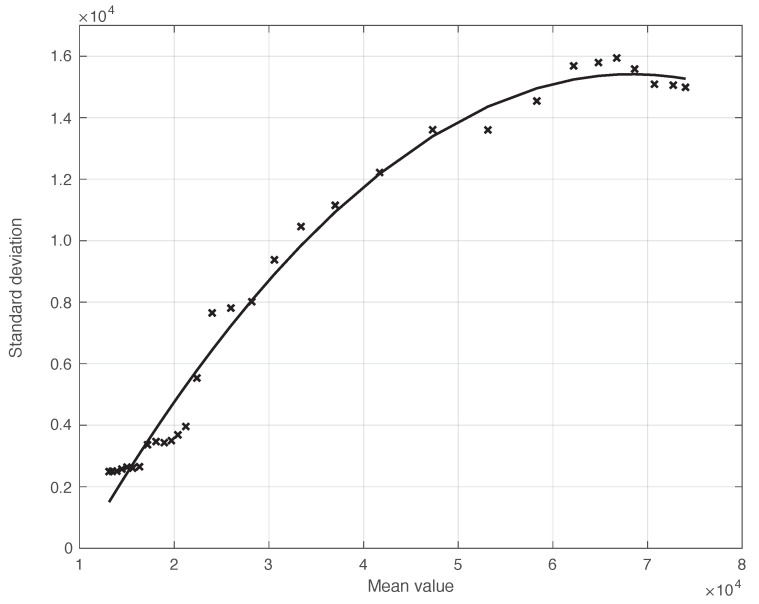
Standard deviation of the local fluctuations of the real SARS-CoV-2 Italian positive test sequence from 7 December 2021 to 6 January 2022, as a function of their expected value. The continuous line represents the best fit with a degree 2 polynomial model.

**Figure 8 vaccines-10-01788-f008:**
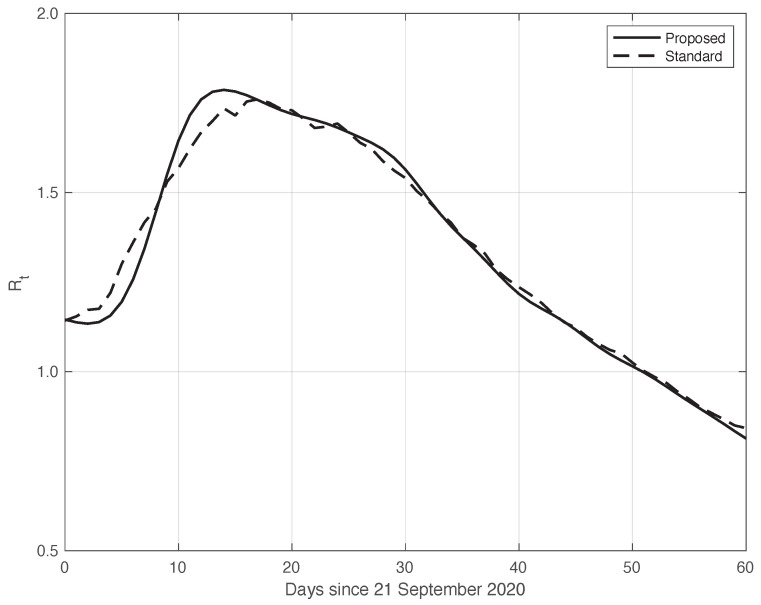
Sequence of the reproduction number $R_{t}$ for real SARS-CoV-2 Italian data from 21 September to 20 November 2020, illustrated in Figure 1. The continuous line corresponds to the proposed method, while the dashed line corresponds to the standard approach.

**Figure 9 vaccines-10-01788-f009:**
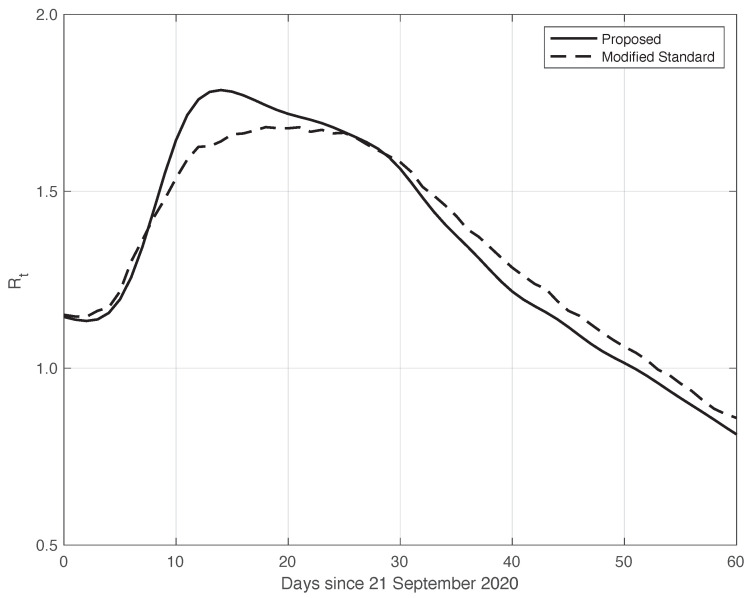
The same as in Figure 8, but the modified standard $R_{t}$ curve is now obtained by first applying estimator (Equation 6) to the data reduced by the systematic error, and then performing arithmetic averaging in time intervals $I_{t}$.

**Figure 10 vaccines-10-01788-f010:**
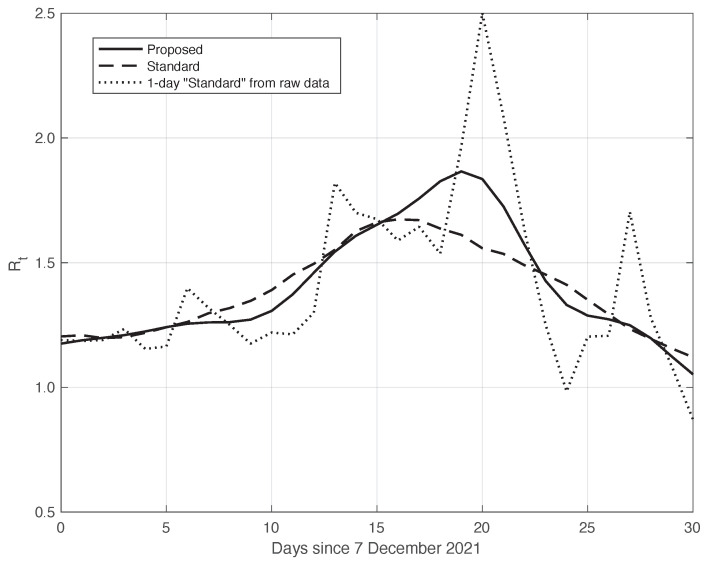
Sequence of the reproduction number $R_{t}$ for real SARS-CoV-2 Italian data from 7 December 2021 to 6 January 2022. The continuous line corresponds to the proposed method, the dashed line to the standard approach, while the dotted line is relative to the estimation by (Equation 6) from the raw data.

**Figure 11 vaccines-10-01788-f011:**
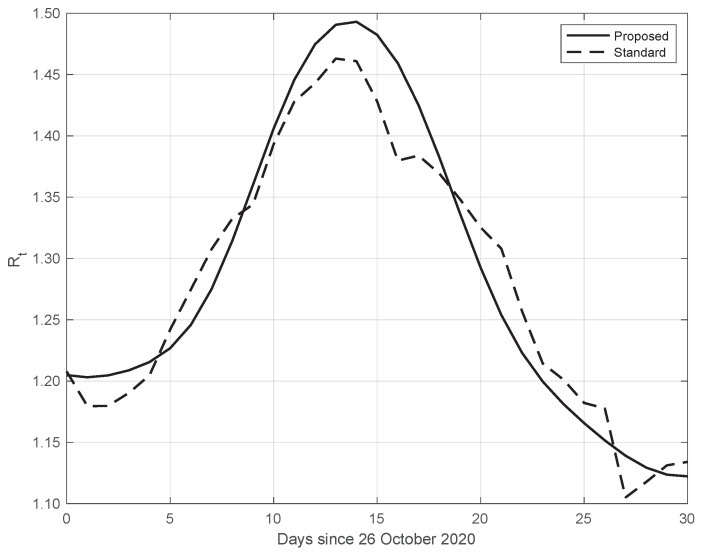
Sequence of the reproduction number $R_{t}$ for real SARS-CoV-2 data in New York city, from 26 October to 25 November 2020. The continuous line corresponds to the proposed method, while the dashed line corresponds to the standard approach.

**Figure 12 vaccines-10-01788-f012:**
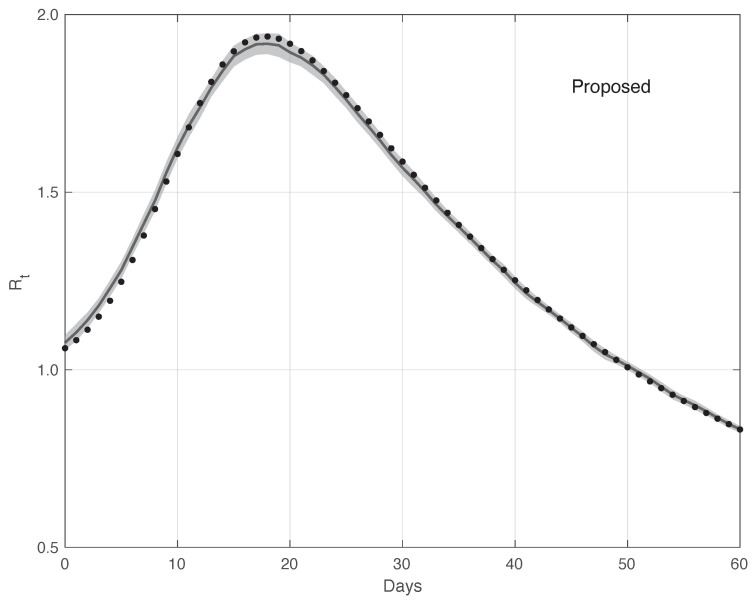
Mean of the $R_{t}$ curve estimated by the proposed method from a realistic SARS-CoV-2 first symptoms synthetic data sample (n=100). The relative 95% CI is shown as a grey shadow. The “true” $R_{t}$ sequence is given as a dotted line.

**Figure 13 vaccines-10-01788-f013:**
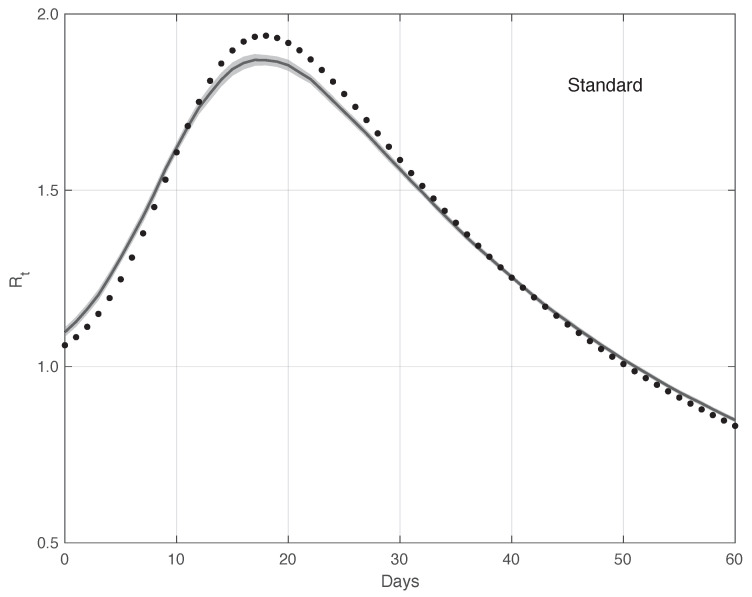
Mean of the $R_{t}$ curve estimated by the standard method from a realistic SARS-CoV-2 first symptoms synthetic data sample (n=100). The relative 95% CI is shown as a grey shadow. The “true” $R_{t}$ sequence is given as a dotted line. The mean is obtained from 100 independent realizations of the synthetic data.

**Figure 14 vaccines-10-01788-f014:**
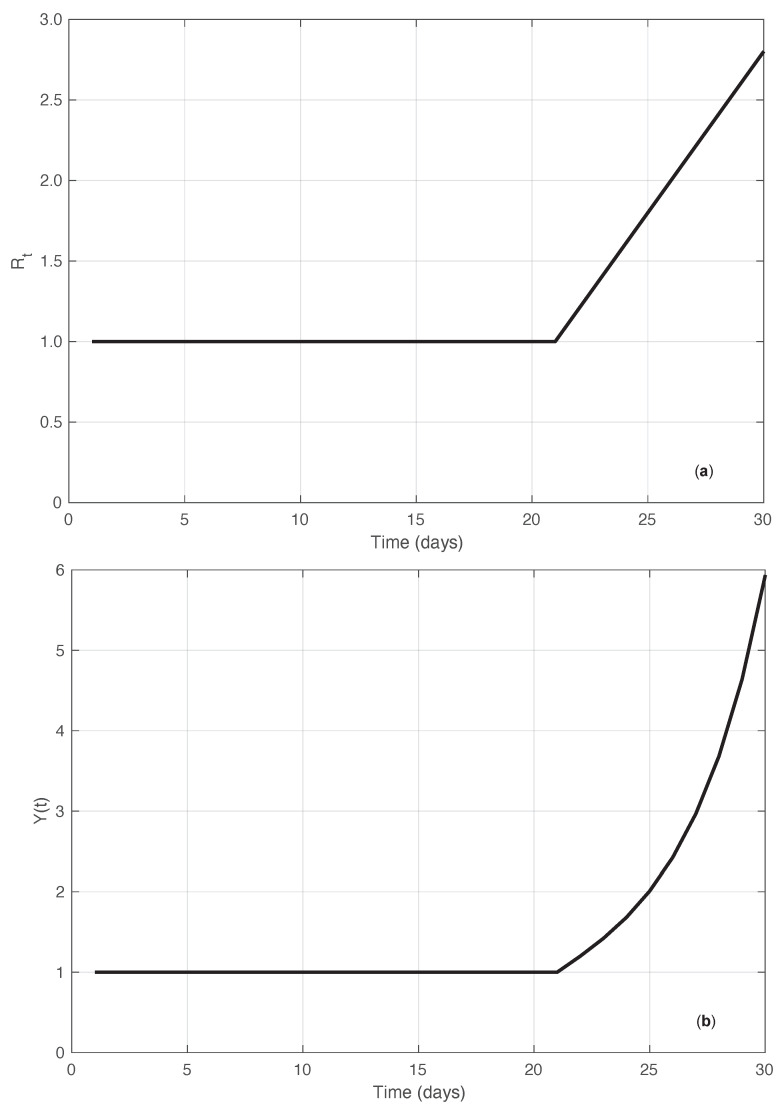
Synthetic epidemic data of the second type (see Section 3.2). In panel (**a**), we show the $R_{t}$ sequence along time. The value of the slope α of the linear growing phase is 0.2 (days${}^{-1}$). In panel (**b**), we show the corresponding incidence data recursively obtained from Equation (Equation 12).

**Figure 15 vaccines-10-01788-f015:**
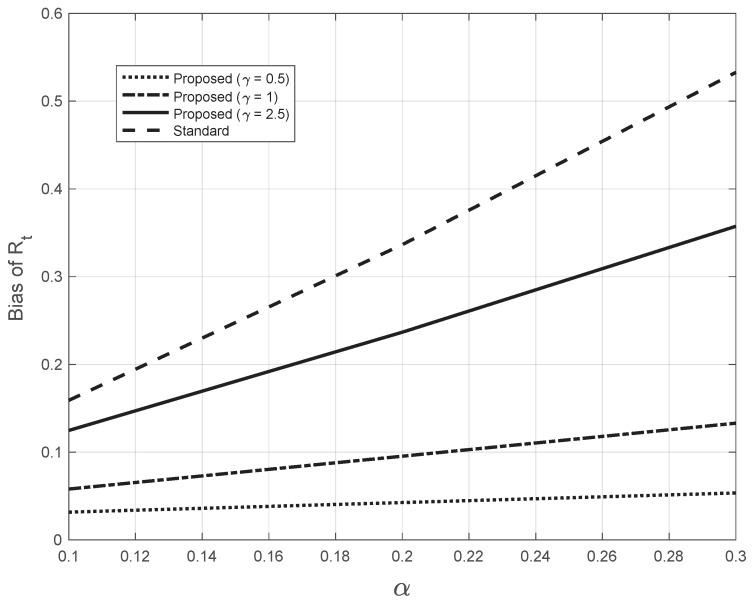
Bias of the $R_{t}$ estimated from the synthetic epidemic data, illustrated in Figure 14. The bias is plotted against the slope α. The dashed line corresponds to the estimate from the standard method. The other lines correspond to the proposed method, with three fixed values for the bandwidth γ in Equations (Equation 7), as indicated in the legend. The last value chosen is close to the one (2.7) found by the proposed method for the real SARS-CoV-2 Italian data from 21 September to 20 November 2020, illustrated in Figure 1.

## Data Availability

Positive test and first symptoms data are available at https://github.com/pcm-dpc/COVID-19/tree/master/dati-regioni and https://www.epicentro.iss.it/coronavirus/sars-cov-2-sorveglianza-dati, respectively.

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
