# Peer review of "New Insights into the Estimation of Reproduction Numbers during an Epidemic"

_vaccines, 2022, doi:10.3390/vaccines10111788_

Round 1

Reviewer 1 Report

The authors study the problem of estimating the reproduction number along an epidemic. They provide a modification of a commonly used method. This approach is interesting and is consistent with a modern approach based on heterogeneity. 

The paper is well written and outlined. I recommend it for publication as is.

Author Response

we are thankful to this Reviewer for her/his feedback. No changes were required.

Reviewer 2 Report

Overall, this is a very hard to understand paper and would need a major rewrite before it
should be accepted for publication. Apart from being rewritten so that it is easier to understand,
the paper is way too long for what it is trying to convey.

The problems start from the very beginning.
* The estimation of the reproduction numbers should be DURING an epidemic, not ALONG an epidemic.
* The use of "ones" should be limited, if not removed. For example, on page 1 line 3,
"we consider the use of positive test data as an alternative to the first symptoms ones that
are typically used." would be better written as "we consider the use of positive test CASE
data as an alternative to the first symptoms DATA, WHICH are typically used." (note that CAPITALS
are used to highlight the changes).
* Page 1 line 11 - remove "some advantages by"
* Page 1 line 11 - replace "sequence" with "data"
* Page 1 line 12 - remove "we propose" and replace "one" with "estimator using first symptoms data"
* Page 1 line 12 - replace "We hope" with "It is hoped"
* Page 1 line 17 - replace "positive test Y_pt and first symptoms Y_fs" with
    "positive test case data (Y_pt) and first symptoms data (Y_fs)"
* Page 1 line 19 - move "respectively" to the end of the sentence.
* Page 1 line 26 - remove "usually" and "drawbacks or"
* Page 1 line 27 - replace "have" with "has" and "referred to" with "associated with"
* Page 1 line 28 - remove "Indeed" and "sampling"
* Page 1 line 29 - replace "day" with "day of the test" and "Besides that" with "In addition"
* Page 1 line 31 - replace "ones" with "tests"
* Page 1 line 33 - replace "these" with "this"
* Page 2 line 36 - remove "always"
* Page 2 line 37 - replace "in the literature, which provides a large number of different
methodologies, approaches and practical applications" with "of research. A large number of
different methodologies, approaches and practical applications have been developed."
* Page 2 line 39 - replace "In the paper by Fraser [3], for example, the author proposes" with
"Fraser [3] proposes"
* Page 2 line 40 - it should be "Kermack-McKendrick" not "Kermack McKendrick"
* Page 2 line 41 - remove "some" and replace "he also" with "In addition, Fraser [3]"
* Page 2 line 42 - what does the term "specific households unit of transmission mean"?
* Page 2 line 43 - replace "for the SARS-CoV-2" with "to the SARS-CoV-2"
* Page 2 line 45 - replace "We cite in this latter regard the paper [7] by Zhao et al, where a"
with "For example, Zhao et al [7]"
* Page 2 line 52 - what is the Italian one you are referring to?
* Page 2 line 54 - remove "as the one in"
* Page 2 line 56 - remove "Besides"
* Page 2 line 58 - remove ", ie regional,"
* Page 2 line 60 - replace "The method in [12]" with "The method described by Cori et al [12]"
* Page 2 line 63 - remove "In that paper"
* Page 2 line 68(?) - remove "as well as others" and "Theoretically"
* Page 2 line 69 - remove "so called" and remove quotes around "serial interval". There should
also be a reference for what the serial interval is.
* Page 2 line 78 - remove "the sake of" and "in both cases"
* Page 2 line 80 - replace "be collected" with "collect"
* Page 3 line 83 - replace "when Rt" with "if Rt" and remove ", in which case the Rt is indeed
the eigenvalue"
* Page 3 line 84 - replace "these eigenfunctons" with "the eigenfunctions"
* Page 3 line 86 - the formula for the Gamma PDF should be on a separate line (as in (2))
* Page 3 line 88 - remove "Gamma" from quotes
* Page 3 line 90 - what are the parameters 1.87 and 3.57? Are they are shape and scale?
* Page 3 line 120 - remove "The paper is organized as follows"
* Page 3 line 124 - remove ", where some conclusions are also drawn."
* Page 4 line 135 - replace "independently and identically distributed as a Poisson" with
"independently and identically distributed (i.i.d.) as a Poisson distribution"
* Page 4 line 140 - why are the shape parameters a=1 and b=5 chosen?
* Page 4 lines 149 and 164 - remove the equal signs
* Page 5 line 171 - what do you mean that they show a periodic pattern? Do you mean the pattern
repeats every 7 days? Or that there is a 7 day lag between first symptoms being recorded
and a positive test?
* Page 6 line 212 - replace "concolving" with "the convolution of"
* Page 6 line 213 - what is mean by a proper kernel?
* Page 14 - be consistent with your labelling. Figure 1a has 8000, 10000, 12000 etc on the y-axis,
but Figure 1b has 1 x 10^4, 1.5 x 10^4 etc. Also, are there the number of cases or the incidence
per thousand/million people?
* Page 17 - add the corrected data for the first symptoms data
* Page 18 - there should be a line representing Rt=1 on the figure, as this represents whether
an epidemic spreads or dies out.
* Page 19 - on the y-axis, use 0.5, 1.0, 1.5, 2.0 not 0.5,1,1.5,2

Reviewer 3 Report

Dear Authors

Well done for the wonderful research.

The authors estimated Rt by using positive test data as an alternative to the first symptoms by studying both theoretically and empirically the relationship between the two approaches.

They then modified a method for estimating Rt along an epidemic. This procedure is not affected by the problems derived from the hypothesis of Rt local constancy assumed in the standard approach. This is  illustrated by the results obtained by applying the proposed methodologies to real and simulated SARS-CoV-2 datasets.

Specific methods are applied  to reduce systematic and random errors affecting the data. The results show that the Rt along an epidemic can be estimated with some advantages by using the positive test sequence, and that the estimator proposed outperforms the standard one. Thus these techniques proposed could help studying and controlling epidemics, in particular the current SARS-CoV-2 pandemic.

Author Response

(The authors gave the same response as above.)

Round 2

Reviewer 2 Report

I think that the authors have address most of the issues raised. There are still a few things that need to be looked at.

1. page 4 line 164 - this sentence should be changed to say that there is a weekly component in the sequence, as opposed to a periodic pattern. For example, the positive test data does not look the same every week, it just goes through the same cycle such as goes up on Monday, down on Tuesday and peaks on Wednesday.

2. page 5 line 174 - change the kernel back to having K(x) on a separate line

3. When referencing two or more works, put a space between them. For example, page 5 line 168 [22, 23] not [22,23]

4. Figure 11 seems out of place with the other figures. Firstly, the numbering should be consistent (1.15, 1.20, 1.25, etc). Secondly, the font size is larger than in other figures.
